# A Case of Gastroallergic and Intestinal Anisakiasis in Italy: Diagnosis Based on Double Endoscopy and Molecular Identification

**DOI:** 10.3390/pathogens12091172

**Published:** 2023-09-18

**Authors:** Stefano D’Amelio, Ilaria Bellini, Claudia Chiovoloni, Cristina Magliocco, Annamaria Pronio, Arianna Di Rocco, Ilaria Pentassuglio, Marco Rosati, Gianluca Russo, Serena Cavallero

**Affiliations:** 1Department of Public Health and Infectious Diseases, Sapienza University of Rome, 00185 Rome, Italy; stefano.damelio@uniroma1.it (S.D.); ilaria.bellini@uniroma1.it (I.B.); claudia.chiovoloni@uniroma1.it (C.C.); ariannadirocco@aslroma2.it (A.D.R.); gianluca.russo@uniroma1.it (G.R.); 2Sandro Pertini Hospital, 00157 Rome, Italy; cristina.magliocco@aslroma2.it (C.M.); ilariapentassuglio@aslroma2.it (I.P.); marcorosati@aslroma2.it (M.R.); 3Digestive Endoscopy Unit, Department of General Surgery and Surgical Specialties “Paride Stefanini”, Sapienza University of Rome, Azienda Policlinico Umberto I, 00161 Rome, Italy; annamaria.pronio@uniroma1.it

**Keywords:** gastroallergic anisakiasis, intestinal anisakiasis, gastric endoscopy, colon endoscopy, molecular diagnosis

## Abstract

Nematodes of the genus *Anisakis* (Rhabditida, Anisakidae) are zoonotic fish-borne parasites and cause anisakiasis, a disease with mild to severe acute or chronic gastrointestinal and allergic symptoms and signs. Anisakiasis can potentially lead to misdiagnosis or delay in diagnosis, and it has been suggested as a risk factor for gastrointestinal tumors. Here, we describe a case report of a 25-year-old woman who presented with gastrointestinal (abdominal pain, nausea, diarrhea) and allergic (diffuse skin rash) symptoms and reported ingestion of raw fish contaminated by worms. Gastro and colon endoscopy allowed the visualization and removal of nematodes and collection of bioptic tissue from ulcers and polyps. The removed nematodes were molecularly identified as *Anisakis pegreffii*. The patient was treated with chlorphenamine maleate, betamethasone, omeprazole, paracetamol, albendazole. We conclude that an upper endoscopy matched with a colonoscopy and molecular characterization of the pathogen yields the most reliable diagnosis and treatment for human anisakiasis, enabling the complete removal of the larvae and preventing chronic inflammation and damage.

## 1. Introduction

Anisakiasis is a fish-borne zoonosis caused by the marine nematode *Anisakis* Dujardin, 1845 spp. (Ascaridoidea: Anisakidae). Infective third-stage larvae (L3) are usually found in the viscera of many teleosts, and they tend to migrate from the abdominal cavity into the fillets, thus representing a zoonotic risk for humans when raw or undercooked fish is consumed. Moreover, *Anisakis* nematodes are able to trigger an allergic reaction [1]. Anisakiasis can manifest in acute (skin rashes, vomiting, diarrhea, anaphylaxis, and severe allergic reactions) and chronic forms (potentially leading to erosive ulcers, granuloma formation, and chronic inflammation) or even asymptomatic. According to the L3 localization and symptoms, the disease can be classified as gastric anisakiasis (GA), gastro-allergic (GAA), intestinal (IA), and extragastrointestinal anisakiasis [2,3,4]. Based on larval behavior and the degree of tissue penetration, invasive larvae are usually found in the mucosa or submucosa of the gastric and intestinal walls, and non-invasive forms can be observed [4]. Although prevalent in Japan, with an estimated annual average incidence of about 20,000 cases [5], the increased popularity of raw fish makes anisakiasis a public health concern for Spain, South Korea, Italy, and the USA, being the top five countries with the highest number of published human anisakiasis cases [6]. However, the global burden of anisakiasis is likely to be severely underestimated, particularly because of the wide repertoire of symptoms that make the diagnosis difficult and the intrinsic limitations of currently available diagnostic tools [7]. For instance, exposure to *Anisakis*-contaminated food is often detected using skin prick test, based on the use of whole body antigen extracts from *Anisakis* L3 or the identification of specific IgE antibodies in patients’ blood. However, both tests are of limited value: false positive results may be produced for cross-reactivity due to the homology between *Anisakis* spp. allergens and antigens of other organisms (i.e., nematodes, shellfish, and arthropods). Moreover, not all patients diagnosed with anisakiasis are positive for serology for specific IgE [4]. For these reasons, endoscopic and X-ray examinations, which may show the presence of worms, edemas, gastric or duodenal ulcers, lesions, and granulomas, are the most reliable diagnostic approaches.

## 2. Case Presentation

A 25-year-old Italian woman was admitted to the emergency room (ER) (Rome, Italy) complaining of the onset of abdominal pain, nausea, and mild diarrhea 5 h following ingestion of marinated anchovies infected with living nematodes. The patient presented a diffuse skin rash on the lower and upper limbs, pubic area, and torso. The anamnesis revealed no history of drug or food allergies. Physical examination revealed tachycardia, while other parameters remained normal. The patient was afebrile with a temperature of 36.7 °C. The blood pressure was 110/65 mmHg, and O_2_ saturation was 99%. A total blood count revealed a high white blood cell count (12.92 × 10^3^ cells/µL). Chemical analyses revealed an elevated C-reactive protein (CRP 1.4 mg/dL), while other parameters remained normal. The patient presented a moderately globose abdomen due to meteorism, with abdominal tenderness on deep palpation in the epigastrium, the left hypochondrium, and the right iliac fossa. The patient declared a pain rate of 5/10. Blumberg, Murphy, and MC Burney’s signs were negative. The electrocardiogram, chest X-ray, and abdominal X-ray results were all normal, except for mild air-fluid levels within the intestine.

On admission to the ER, the patient was administered chlorphenamine maleate (10 mg/100 mL), betamethasone (4 mg as a single bolus injection), omeprazole (40 mg as a single bolus injection), paracetamol (1000 mg/100 mL) and 500 mL saline solution in consideration of the anaphylactic event.

The anamnesis was suggestive of a fish-borne zoonosis transmitted by nematodes; therefore, the National Center for *Anisakis* infections in Italy and the Parasitology Section of the Department of Public Health and Infectious Diseases (DSPMI) of the Sapienza University of Rome was contacted by the ER personnel.

A double endoscopic approach (an esophagogastroduodenoscopy EGD and a colonoscopy) was carried out.

The EGD examination revealed five larval nematodes in multiple locations of the gastric lumen: four in the gastric fundus, the lesser curvature, and the gastroesophageal junction. One larva was localized in the gastric antrum near an erosive lesion. All larvae were successfully recovered using endoscopic forceps (Figure 1) and sent to the DSPMI. Biopsies of the gastric and duodenal mucosa where the larvae were located were performed.

The ileo-colonoscopy revealed a sessile polyp (6 mm) and an acute mucosal ulcer (6 mm) in the ascending colon. The polypoid lesion was removed with hot snare polypectomy, and biopsies of the ulcer were performed. Two larval nematodes were also removed from the transverse colon using endoscopic forceps and sent to the DSPMI (Figure 2).

Histological examinations of the biopsy of the gastric fundus and antrum revealed non-atrophic gastritis (also referred to as chronic superficial). Histological sections of the ascending colon stained with hematoxylin and eosin revealed inflammatory infiltrate of eosinophils in the crypts of Lieberkühn of the mucosal layer (Figure 3). Infiltration of eosinophils together with leukocytes covering the ulcers was reported. There are few to no signs of chronic phlogosis occurring in the submucosal layer.

Histological examination of the removed colonic polyp showed a sessile serrated adenoma (SSA).

The larval nematodes recovered during endoscopies were sent to the DSPMI of Sapienza University of Rome for molecular identification of species. Nematodes were observed under a light microscope for gross morphological identification, and some typical morphological traits corresponding to *Anisakis* type I larvae were present, as the presence of a larval tooth at the apical end (lips), the shape of esophageal ventriculus, anal glands, and distinct mucron at the end of the tail (Figure 4).

Genomic DNA was isolated using the Isolate II^®^ Genomic DNA purification kit (Bioline). Amplicons were generated using primers NC5 (forward: 5′-GTAGGTGAACCTGCGGAAGGATCAT-3′) and NC2 (reverse: 5′-TTAGTTTCTTCCTCCGCT-3′) targeting nuclear rDNA-ITS (internal transcribed spacers 1, 5.8 S and internal transcribed spacers 2 rRNA gene), according to D’Amelio et al., 2000 [8]. Then, positive ITS amplicons were digested for 4 h at 37 °C using the restriction enzyme *HinfI*, which produced three fragments of ~370, 300, and 250 bp. Based on the RFLP pattern obtained (Figure 5), the parasites were identified as *Anisakis pegreffii* Campana-Rouget & Biocca, 1955 [8].

The patient’s health status improved shortly after removal of the worms. Following the diagnosis of an allergic reaction complicated by acute gastric anisakiasis, albendazole (400 mg/1 cp) was added to the therapeutic regimen. Albendazole (400 mg/1 cp), betamethasone (1 mg/1 cp × 5 days, then ½ cp × 3 days), and bilastine (20 mg/1 cp for 5 days) were prescribed after hospital discharge. A medical check-up of the patient carried out one day after hospital discharge confirmed no recurrence of her symptoms.

## 3. Discussion

The first case of anisakiasis in an Italian patient was described in 1996 by Stallone et al. [9], with steadily rising reports ever since [10]. The main etiological agent of anisakiasis in Italy is *A. pegreffii*, as suggested in previous records molecularly identified [11], although the ingestion of infected imported fish can determine diseases due to other anisakid species, such as *Pseudoterranova decipiens* sensu stricto Paggi L, Mattiucci S, Gibson DI, Berland B, Nascetti G, Cianchi R, Bullini L., 2000, from other geographic areas [12]. Nowadays, the prevalence of human anisakiasis is believed to be significantly underestimated due to the limitations of available diagnostic tools and the poor specificity of the clinical symptomatology. Since routine laboratory tests are non-specific, a thorough anamnesis is key to discovering exposure to *Anisakis*-contaminated food. Patients could present with elevated acute-phase reactants, mild or no leukocytosis, and may or may not show an allergic reaction.

In the present case, the patient presented with acute abdomen and epigastric pain within 1 to 3 days from exposure, typical signs of GA, classically considered predominant with respect to the IA form [13,14], with concomitant skin rash and hives, matching the description of GAA [15]. However, while an elevated white blood cell count and eosinophilia can assist with the diagnosis, no peripheral eosinophilia upon hospitalization was detected in the patient, as frequently reported in similar clinical cases [16].

As previously mentioned, the gastrointestinal wall of a patient affected by anisakiasis typically exhibits different phases, such as an eosinophilic phlegmon, a massive eosinophil infiltration, and marked edema throughout the submucosal space of the small intestine, arising shortly after the penetration of the larva [17], and a subsequent abscess and/or a granuloma formation at histological analyses [18]. On endoscopic examination and histology, the patient of the present case report showed mucosal edema and ulcers mediated by eosinophilic infiltration and degranulation [14], consistently with the *Anisakis* infection. If ulcers are normally associated with the acute gastric manifestation of the disease [19], recent case reports showed the occurrence of chronic anisakiasis and localized tumors of the submucosal layer, characterized by eosinophilic granuloma and embedded dead *Anisakis* larvae [20]. Interestingly, even though the mucosa showed no sign of chronic phlogosis here, which could represent a risk factor for neoplastic transformations, a polyp identified as an SSA was observed within the colon. SSAs are recognized precursors to microsatellite unstable adenocarcinomas, responsible for up to 20% of colorectal carcinomas [21].

Leading clinical references on the treatment of infectious diseases and anti-infective drugs administration established the endoscopic removal of larvae as the treatment of choice for anisakiasis, which leads to both diagnosis and treatment when worms are removed with endoscopic forceps and identified [22]. Despite the fact that a single larva is enough to trigger the clinical manifestations of the disease, contaminated food may present more than one larva. This is relevant for the diagnosis and the treatment of the patient, considering that about 40% of colonic anisakiasis are estimated to be asymptomatic, and colonoscopy is not usually performed in the absence of intestinal symptoms [13]. However, *Anisakis* larvae may still migrate towards the bowel, causing IA after a gastric involvement [4]. Therefore, in our opinion, to avoid missing asymptomatic colonic anisakiasis, an upper endoscopy matched with a colonoscopy yields the best diagnosis and treatment for human anisakiasis in order to remove all larvae inside the host and avoid chronic inflammation and damage. Since a retrospective study suggested an association between anisakiasis and an increased risk of developing gastrointestinal tumors [23] and case reports described the incidental finding of *Anisakis* larvae at the gastrointestinal tumor site [24,25,26], regular check-ups following the removal of the larvae should be considered. Even though the survival time of *Anisakis* in humans (accidental host) is limited, if no colonoscopy is performed and all larvae are not completely removed, inflammation can exacerbate through the infiltration of eosinophils and the proliferation of connective tissue around the larval body [4].

To ensure complete removal after endoscopic extraction, there are reports of using albendazole 400 mg orally once or twice a day, from three to several days [22,27], in line with WHO-approved recommended doses for other helminthic infections [28]. Antihelminthic regimes for anisakiasis are listed as discretional or rarely necessary [28], and data on their efficacy is limited. Following the double endoscopy and the albendazole-based regime, the patient reported the complete resolution of the symptoms, as referred to during the medical check-up one day after hospital discharge. However, long-term follow-up examinations of outpatients should be contemplated in view of the possibility of asymptomatic cases of anisakiasis and its possible correlation with medically significant sequelae. Patients who experience allergic reactions should be informed to avoid the ingestion of marine seafood that may contain *Anisakis*.

## 4. Conclusions

In conclusion, if anisakiasis is suspected in a patient with gastrointestinal and allergic signs and symptoms, double endoscopy (gastric and colon endoscopy) is the decisive diagnostic and therapeutic approach for the efficient resolution of the disease. Moreover, according to the rising case reports in the scientific literature describing the co-occurrence of anisakiasis and tumors, long-term medical surveillance should be envisaged.

## Figures and Tables

**Figure 1 pathogens-12-01172-f001:**
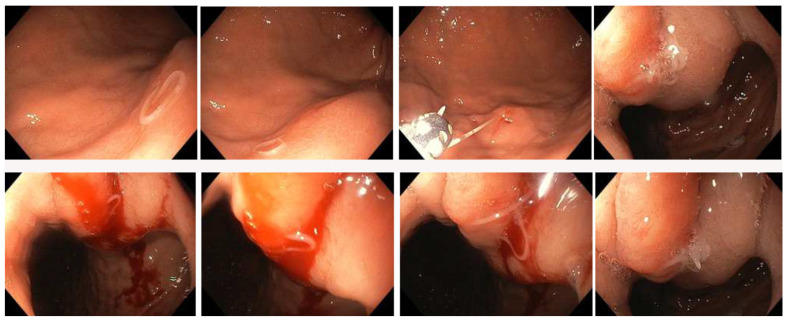
Gastric endoscopy. Endoscopy of the gastric portion of the digestive system with *Anisakis* larvae visible.

**Figure 2 pathogens-12-01172-f002:**
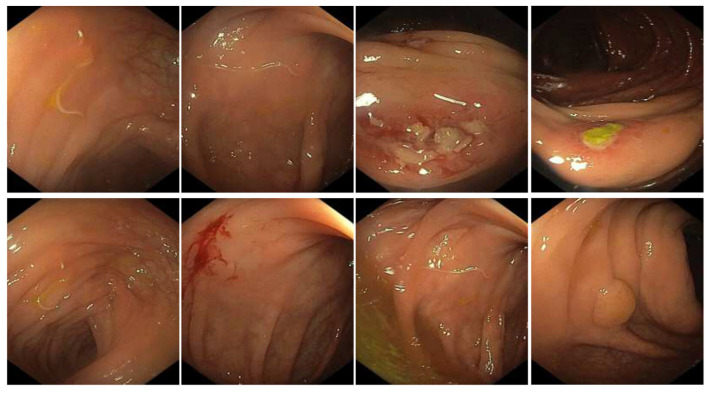
Colon endoscopy. Endoscopy of the intestinal portion of the digestive system with polyp (6 mm), ulcer (6 mm), bleeding, and *Anisakis* larvae visible.

**Figure 3 pathogens-12-01172-f003:**
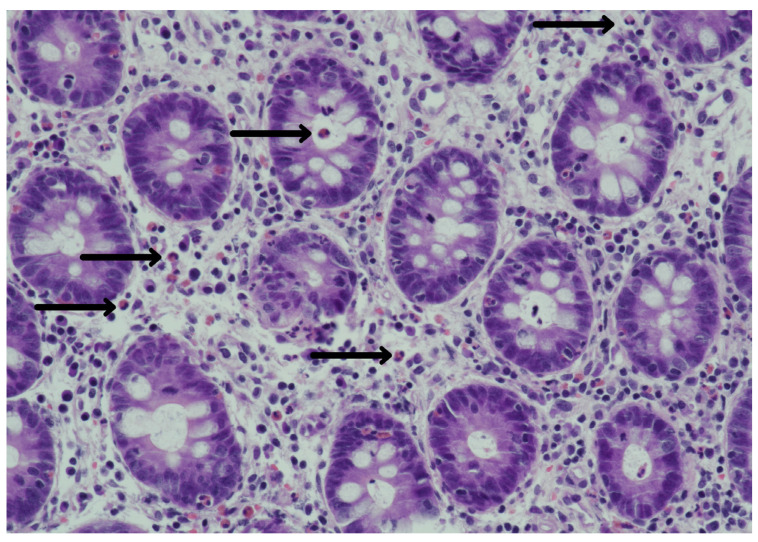
Histology of intestinal eosinophilic cryptitis. Intestinal eosinophilic cryptitis, with infiltration of the mucosal layer of the ascending colon, as indicated by the black arrows.

**Figure 4 pathogens-12-01172-f004:**
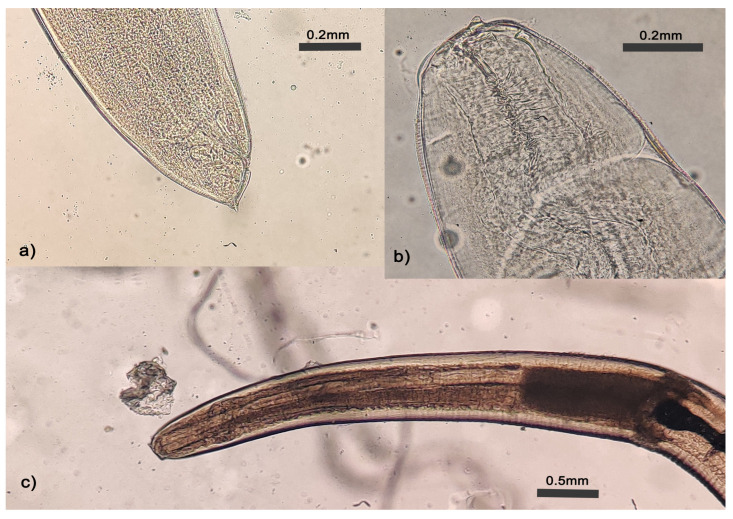
*Anisakis* type I larvae. Morphological features corresponding to *Anisakis* larvae Type I were observed under a light microscope. (**a**) tail with mucron (most caudal part of the tail). (**b**). Buccal lips with larval tooth (triangular-shaped); (**c**) L3 with visible digestive tract and esophageal ventriculus (darker shade) of Type I larvae. Scale bar is indicated.

**Figure 5 pathogens-12-01172-f005:**
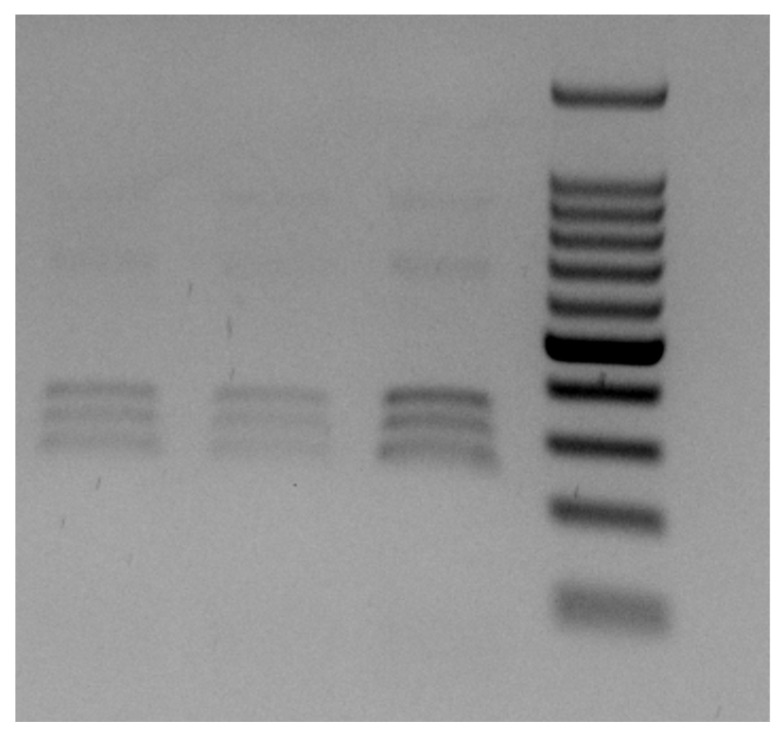
PCR-RFLP of ITS from *Anisakis pegreffii*. Banding pattern obtained after PCR-RFLP of the ITS of *A. pegreffii* L3 recovered in the study. Three representative specimens are visible, with the molecular ladder (100 bp ladder with central intense band corresponding to 500 bp).

## Data Availability

Data created are available in the publication. No further data is available in additional repositories.

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
