# Peer review of "A Case of Gastroallergic and Intestinal Anisakiasis in Italy: Diagnosis Based on Double Endoscopy and Molecular Identification"

_pathogens, 2023, doi:10.3390/pathogens12091172_

Round 1

Reviewer 1 Report

Anisakiasis remains a public health problem in many Mediterranean coastal countries. Stefano D’Amelio with coauthors presented us with a case of anisakiasis in Italy. This manuscript makes a certain contribution to our knowledge about reliable diagnostics and treatment for human anisakiasis. The authors give recommendations for the worm prevention of anisakiasis.

I have some minor remarks about this manuscript.

1. Since this is a Сase Report, I advise the authors to use the traditional sections: Introduction, Case Presentation (Former Methods and Results), Discussion and Conclusion.

2. I propose to change the first sentence of the abstract a little: “Nematodes of the genus Anisakis (Rhabditida, Anisakidae) are the zoonotic fish-borne parasites and causes anisakiasis …”

3. Line 35 – “Moreover, Anisakis nematodes are able to …”

4. In figures 3 and 4, need to show the scale bars.

5. According International Code of Zoological Nomenclature (ICZN) at the first mention of genera and species (as in lines 134, 155) its full Latin name with the author and year of description should be given; in relation all species of living organisms (Anisakis Dujardin, 1845; Anisakis pegreffii Campana-Rouget & Biocca, 1955; Pseudoterranova decipiens (Krabbe, 1878)). On subsequent mentions, the generic name is abbreviated (as in line 153 – A. pegreffii, etc.).

6. Line 173 – Please use “infection” instead “infestation” here.

7. Discussion missing paragraph indents.

8. Please, correct reference in line 131 – D’Amelio et al. [number]. There is no similar article from 2000 in the list of references. Probably you meant No 10 from 1999?

Author Response

Dear reviewer, thanks for the efforts in revising the case report we have submitted. We addressed all the raised issues with the exception of one comment (see point 4) hoping that the document is improved. Please find a detailed rebuttal below, on the behalf of all authors.

D’Amelio with coauthors presented us with a case of anisakiasis in Italy. This manuscript makes a certain contribution to our knowledge about reliable diagnostics and treatment for human anisakiasis. The authors give recommendations for the worm prevention of anisakiasis.

I have some minor remarks about this manuscript.

  1. Since this is a Ð¡ase Report, I advise the authors to use the traditional sections: Introduction, Case Presentation (Former Methods and Results), Discussion and Conclusion.

Reply: we thank the reviewer and considering comments from the reviewer 3 as well, we have changed the paragraph title as such: Introduction, Case Presentation, Discussion

  1. I propose to change the first sentence of the abstract a little: “Nematodes of the genus Anisakis(Rhabditida, Anisakidae) are the zoonotic fish-borne parasites and causes anisakiasis …”

Reply: we have changed the sentence as suggested.

  1. Line 35 – “Moreover, Anisakisnematodes are able to …”

Reply: we have changed the sentence as suggested.

  1. In figures 3 and 4, need to show the scale bars.

Reply: we thank the reviewer and we have added the suggested scale bar for figures 4, as it can be informative for the identification of the worm. We did not include the scale bar on figure 3 because it is not informative for the measurements or identification of intestinal cells (the dimensions of eosinophils and crypts of Lieberkhun are known).  

  1. According International Code of Zoological Nomenclature (ICZN) at the first mention of genera and species (as in lines 134, 155) its full Latin name with the author and year of description should be given; in relation all species of living organisms (AnisakisDujardin, 1845; Anisakis pegreffiiCampana-Rouget & Biocca, 1955; Pseudoterranova decipiens (Krabbe, 1878)). On subsequent mentions, the generic name is abbreviated (as in line 153 – A. pegreffii, etc.).

Reply: we thank the reviewer and we have added the suggested nomenclature. In particular, Anisakis Dujardin, 1845 in the very first line of the introduction and Anisakis pegreffii Campana-Rouget & Biocca, 1955 after the molecular identification and Paggi L, Mattiucci S, Gibson DI, Berland B, Nascetti G, Cianchi R, Bullini L. 2000 after the species mention of P. decipiens s.s.

  1. Line 173 – Please use “infection” instead “infestation” here.

Reply: we have changed the term accordingly.

  1. Discussion missing paragraph indents.

Reply: done.

  1. Please, correct reference in line 131 – D’Amelio et al. [number]. There is no similar article from 2000 in the list of references. Probably you meant No 10 from 1999?

Reply: we thank the reviewer and we have included also the missing ref about D’Amelio et al 2000 and changed all the remaining numbers of the references list.

Reviewer 2 Report

Anisakiasis is an important disease, affecting humans. The consumption of raw or insufficiently cooked fish is the main factor for the increasing of the human clinical cases. As mentioned in the text, the number of cases appears to be underestimated, due to the ambiguity of symptoms and difficulty in diagnostic methods. The lack of notification of positive cases by the clinicians is another important factor to be considered.

The methodology is well described and appropriate for this kind or article. The illustrative clinical case is well documented and discussed, resulting in well-founded conclusions. The figures have a good quality and are a good complement for the article. The article is well written, well-structured and easy to understand.

Nevertheless, there are some aspects that should be clarified, valuing the work:

-          Line 59 – Should say “emergency room (ER)”, once you mentioned “ER”at the line 73 “On admission to the ER”;

-          Line 131 – you mentioned that the amplicons were digested “using the restriction enzyme HinfI”, but you didn’t present the procedure conditions;

1.      At the same paragraph, line 133, says that “the parasites were identified as Anisakis pegreffii”. Once the amplicon was not sequenced, your result is based on RFLP, it is important that you find a reference to support your conclusion;   

-          Line 13 – “et al 2000” should be “et al., 2000”;

-          Line 151- should be “et al” (italic).

Author Response

Dear reviewer, thanks for the efforts in revising the case report we have submitted. We addressed all the raised issues hoping that the document is improved. Please find a detailed rebuttal below, on the behalf of all authors.

Anisakiasis is an important disease, affecting humans. The consumption of raw or insufficiently cooked fish is the main factor for the increasing of the human clinical cases. As mentioned in the text, the number of cases appears to be underestimated, due to the ambiguity of symptoms and difficulty in diagnostic methods. The lack of notification of positive cases by the clinicians is another important factor to be considered.

The methodology is well described and appropriate for this kind or article. The illustrative clinical case is well documented and discussed, resulting in well-founded conclusions. The figures have a good quality and are a good complement for the article. The article is well written, well-structured and easy to understand.

Nevertheless, there are some aspects that should be clarified, valuing the work:

-          Line 59 – Should say “emergency room (ER)”, once you mentioned “ER”at the line 73 “On admission to the ER”;

Reply: we thank the reviewer and we have included the suggested abbreviation.

-          Line 131 – you mentioned that the amplicons were digested “using the restriction enzyme HinfI”, but you didn’t present the procedure conditions;

Reply: we have included the experimental conditions “for 4h at 37°C”

  1. At the same paragraph, line 133, says that “the parasites were identified as Anisakis pegreffii”. Once the amplicon was not sequenced, your result is based on RFLP, it is important that you find a reference to support your conclusion;   

Reply: we have added the references number 8 after the sentence of species identification.

-          Line 13 – “et al 2000” should be “et al., 2000”;

Reply: done.

-          Line 151- should be “et al” (italic).

Reply: done.

Reviewer 3 Report

In the present manuscript, the authors described a case report of a 25-year-old woman who presented with gastrointestinal and allergic symptoms and reported eating raw fish. Gastro and colon endoscopy allowed visualization and removal of nematodes and bioptic tissue collection from ulcers and polyps. The removed nematodes were molecularly identified as Anisakis pegreffii. The manuscript is interesting and well written; however, the authors should separate conclusions from discussion.

Some minor comments:

Please write in lowercase the generic name of the medicines administered (lines 22-23, 73-75, 144-146).

Line 22: “Paracetamol, Albendazole” change to “paracetamol, and albendazole”.

Line 43: “20.000” change to “20,000”.

Line 58: “Methods and Results” change to “Case Description”.

Line 59: “emergency room”, please add “(ER)”.

Lines 87-88: please rewrite the sentences, for example: Biopsies of the gastric and duodenal mucosa where the larvae were located were performed.

Line 113: “SSAs” change to “SSA”.

Line 117: “type 1 larvae” change to “type I larvae”.

Line 131: please delete link in “rRNA”.

Line 150: “Discussions and conclusions” change to “Discussion”.

Line 153: Anisakis pegreffii” change to A. pegreffii”.

Line 155: “Pseudoterranova decipiens s.s.” change to “Pseudoterranova decipiens sensu stricto”.

Line 179: “sessile serrated adenoma” change to “SSA”.

Minor editing of English language required.

Author Response

Dear reviewer, thanks for the efforts in revising the case report we have submitted. We addressed all the raised issues hoping that the document is improved. Please find a detailed rebuttal below, on the behalf of all authors.

In the present manuscript, the authors described a case report of a 25-year-old woman who presented with gastrointestinal and allergic symptoms and reported eating raw fish. Gastro and colon endoscopy allowed visualization and removal of nematodes and bioptic tissue collection from ulcers and polyps. The removed nematodes were molecularly identified as Anisakis pegreffii. The manuscript is interesting and well written; however, the authors should separate conclusions from discussion.

Some minor comments:

Please write in lowercase the generic name of the medicines administered (lines 22-23, 73-75, 144-146).

Reply: done.

Line 22: “Paracetamol, Albendazole” change to “paracetamol, and albendazole”.

Reply: done.

Line 43: “20.000” change to “20,000”.

Reply: done.

Line 58: “Methods and Results” change to “Case Description”.

Reply: considering also comments from reviewer 1, we have changed the paragraph title to “Case Presentation”.

Line 59: “emergency room”, please add “(ER)”.

Reply: we have changed the sentence as suggested.

Lines 87-88: please rewrite the sentences, for example: Biopsies of the gastric and duodenal mucosa where the larvae were located were performed.

Reply: we thank the reviewer and we have changed the sentence accordingly.

Line 113: “SSAs” change to “SSA”.

Reply: done.

Line 117: “type 1 larvae” change to “type I larvae”.

Reply: done.

Line 131: please delete link in “rRNA”.

Reply: done.

Line 150: “Discussions and conclusions” change to “Discussion”.

Reply: we have changed the paragraph title as suggested.

Line 153: “Anisakis pegreffii” change to “A. pegreffii”.

Reply: we thank the reviewer and we have included the suggested abbreviation.

Line 155: “Pseudoterranova decipiens s.s.” change to “Pseudoterranova decipiens sensu stricto”.

Reply: we have changed as suggested.

Line 179: “sessile serrated adenoma” change to “SSA”.

Reply: we thank the reviewer and we have included the suggested abbreviation.

Round 2

Reviewer 3 Report

The authors have addressed all my concerns; however, they should consider separating section 3 into “Discussion” and “Conclusions”.

Author Response

Rebuttal to the second round of the revision by the reviewer 3

We thank the reviewer and we have modified the paragraph accordingly, including paragraph 4. Conclusions.

Comments and Suggestions for Authors: The authors have addressed all my concerns; however, they should consider separating section 3 into “Discussion” and “Conclusions”.

Reply: a paragraph 4. Conclusions has been added as follows: “In conclusion, if anisakiasis is suspected in a patient with gastrointestinal and allergic signs and symptoms, double endoscopy (gastric and colon endoscopy) is the decisive diagnostic and therapeutic approach for the efficient resolution of the disease. Moreover, according to the rising case reports in the scientific literature describing the co-occurrence of anisakiasis and tumors, long-term medical surveillance should be envisaged.”